# Detection of Hydrogen Sulfide in Sewer Using an Erbium-Doped Fiber Amplified Diode Laser and a Gold-Plated Photoacoustic Cell

**DOI:** 10.3390/molecules27196505

**Published:** 2022-10-01

**Authors:** Chaofan Feng, Marilena Giglio, Biao Li, Angelo Sampaolo, Pietro Patimisco, Vincenzo Spagnolo, Lei Dong, Hongpeng Wu

**Affiliations:** 1State Key Laboratory of Quantum Optics and Quantum Optics Devices, Institute of Laser Spectroscopy, Shanxi University, Taiyuan 030006, China; 2Collaborative Innovation Center of Extreme Optics, Shanxi University, Taiyuan 030006, China; 3PolySense Lab—Dipartimento Interateneo di Fisica, University and Politecnico of Bari, Amendola 173, 70126 Bari, Italy; 4PolySense Innovations Srl, Amendola 173, 70126 Bari, Italy

**Keywords:** hydrogen sulfide, erbium-doped fiber amplifier, gold-plated photoacoustic cell, sewer gas detection

## Abstract

A photoacoustic detection module based on a gold-plated photoacoustic cell was reported in this manuscript to measure hydrogen sulfide (H_2_S) gas in sewers. A 1582 nm distributed feedback (DFB) diode laser was employed as the excitation light source of the photoacoustic sensor. Operating pressure within the photoacoustic cell and laser modulation depth were optimized at room temperature, and the long-term stability of the photoacoustic sensor system was analyzed by an Allan-Werle deviation analysis. Experimental results showed that under atmospheric pressure and room temperature conditions, the photoacoustic detection module exhibits a sensitivity of 11.39 μV/ppm of H_2_S and can reach a minimum detection limit (1σ) of 140 ppb of H_2_S with an integration time of 1 s. The sensor was tested for in-field measurements by sampling gas in the sewer near the Shanxi University canteen: levels of H_2_S of 81.5 ppm were measured, below the 100 ppm limit reported by the Chinese sewer bidding document.

## 1. Introduction

The urban sewage pipe system is an important infrastructure in cities. It plays a very important role in the construction and development of towns, the improvement of people’s living standards, the protection of people’s health and the urban ecological environment. Under the influence of years of corrosion of sewage, sewer gases, and other external factors, they are extremely prone to accidents such as damage, ground collapse and sewage leakage, causing environmental pollution. For example, in the United States, sewer corrosion is estimated to cost USD 14 billion annually, and this cost will continue to increase over time [1]. The economic impact of sewer corrosion has forced the government to procure a sensor to monitor the concentration of corrosive gases, which would allow for the timely cleaning of sewers before they reach a threshold, preventing further deterioration.

The organic matter in sewers will produce a variety of gases under the action of microorganisms, such as H_2_S, ammonia (NH_3_), and methane (CH_4_) etc. These gases have a great impact on public facilities and residents’ lives. Among these sewer gases, H_2_S gas attracts attention due to its odor of rotten egg and toxicity [2,3]. H_2_S dissolves and produces acidic substances that corrode public facilities. H_2_S is a strong neurotoxin and has a strong stimulating effect on the mucous membrane [4]. The inhalation of H_2_S with a concentration higher than 120 parts per million (ppm) can cause acute poisoning and even death, and low concentrations of H_2_S can also affect the eyes, respiratory system, and central nervous system [5].

One of the main factors affecting the corrosion of sewage pipes is the oxidation of H_2_S in chemical and biological processes. In the sewage pipe system, microbial sulfate-reducing bacteria use various organic matter or hydrocarbons to reduce sulfate, and directly forms H_2_S under the action of alienation [1]. In this process, part of the H_2_S reacts with the water in the sewer to form sulfuric acid and adheres to the sewer wall, corroding the sewer system [6]. Due to these characteristics, this poses challenges for sensors used in H_2_S detection, and it is necessary to ensure that the core components of the sensor have the characteristics of corrosion resistance and long-term stability.

At present, there are many methods for trace gas detection, including gas chromatography [7], semiconductor sensors [8], electrochemical sensors [9], fluorescent probes [10], and chemiluminescence [11] etc. However, gas chromatography has a high cost, a large volume and a relatively long response time, which is not conducive to long-term online detection [12]. Semiconductor sensors are the most widely used gas sensors at present. However, this kind of sensor is greatly affected by the surrounding environment and has higher requirements on temperature and humidity near the measuring site [13]. Electrochemical sensors have low power consumption and compact size, but poor gas selectivity and short service life [14]. The chemiluminescence method is easy to operate and has high sensitivity, but different gases have cross-interference, and the detection cost is high [15].

In recent years, laser absorption spectroscopy (LAS) has attracted more and more attention. In laser absorption spectroscopy the target gas molecules are excited from the ground state to higher energy levels by laser light absorption. The gas molecule can return to the ground state either via radiative processes, corresponding to the emission of photons belonging to the fluorescence or phosphorescence spectrum, or via non-radiative processes, i.e., by collisions with the surrounding molecules in the gas matrix. Among the LAS-based techniques, photoacoustic spectroscopy (PAS) consists in modulating the laser light emission. The consecutive expansion and contraction of the gas sample relaxing energy via non-radiative modes generate heat waves, i.e., sound waves. By detecting these sound wave signals with a microphone, the concentration of the target gas can be deduced [16,17,18,19,20,21]. Since the photoacoustic spectroscopy measures the acoustic signal generated by target gas absorbing the laser energy, PAS is an ideal zero-background detection technique, limited by the thermal noise level. Moreover, acoustic sensors have no wavelength selectivity. Therefore, PAS-based gas sensors can use excitation light sources of various wavelengths from ultraviolet to visible, near-infrared, mid-infrared, and even tera-Hertz light, while employing the same microphone. Finally, photoacoustic spectroscopy gas sensors have small size, fast response time, and high selectivity and sensitivity, which is beneficial to practical applications. Bonilla-Manrique et al. developed an H_2_S photoacoustic sensor based on a 3D-printed resonant cell made of acrylonitrile butadiene styrene, to demonstrate the potential of the use of plastic gas cells for corrosive gas detection [22]. Here, H_2_S gas detection limits of 3 ppm and 0.3 ppm were achieved with a lock-in integration time of 300 ms and 30 s, respectively, employing a mid-infrared external cavity quantum cascade laser (EC-QCL) as the excitation source. A near-infrared diode laser (DL) was instead employed by Wu et al. to develop an off-axis quartz-enhanced PAS H_2_S sensor [3]. Compared to EC-QCLs, DLs are cheaper and do not suffer from grating mechanical instabilities possibly affecting the alignment or laser stability. The detection limits of 734 ppb and 200 ppb were obtained at 1 s and 30 s lock-in integration time. However, the electrical modulation cancellation method was needed to remove the sensor floor noise offset. Moreover, the tuning fork is more complicated in calibrating the optical path. Standard PAS and a near-infrared DL was employed by Yin et al. to detect H_2_S in sulfur hexafluoride buffer [23]. When sulfur hexafluoride was used as a carrier gas, the detection limit reached a level of 109 ppb for an averaging time of 1 s. However, only H_2_S gas under sulfur hexafluoride was measured, and H_2_S gas under nitrogen was not studied.

In this paper, we developed a photoacoustic sensor based on a cheap near-infrared distributed feedback (DFB) diode laser emitting at 1.58 um, used as excitation light source for the detection of H_2_S in sewers. An erbium-doped fiber amplifier (EDFA) was employed to boost the power up to 1.4 W and improve the detection limits. A gold-plated photoacoustic cell was designed and developed, to prevent any corrosion phenomena due to the interaction of H_2_S with the cell. The sensing performance was studied in detail using nitrogen as carrier. A detection limit of 140 parts per billion under 1 s integration time was achieved. Then, the sensor was demonstrated in-field, by analyzing a gas sample collected from the sewer.

## 2. Results and Discussion

In the sewer, together with H_2_S there are many other gases, such as methane (CH_4_), ammonia (NH_3_), carbon dioxide (CO_2_), carbon monoxide (CO), and water (H_2_O) etc. A simulation of the absorption spectrum of these gas species within the DFB emission range is shown in Figure 1, according to the *HITRAN* database. The simulation has been performed at the concentrations reported by the Chinese sewer bidding document and at two different pressures: 100 Torr, and 680 Torr [24].

As shown in Figure 1, high-intensity H_2_S absorption lines can be targeted around 6320.60 cm^−1^. Therefore, with the central wavelength set at 6320.60 cm^−1^, the ramp wave amplitude was set in order to scan the DFB laser emission from 6320.20 cm^−1^ to 6320.95 cm^−1^. It is worth noting that the signal amplitude of the photoacoustic sensing system is related to the absorption coefficient of the target gas absorption line (see Equation (1)). As the pressure changes, the gas absorption linewidth changes and the absorption coefficient will change as well, as shown in Figure 1. Moreover, the pressure-related broadening of the absorption features causes absorption lines merging at high pressure. For example, the three separate features peaked at 6320.38 cm^−1^, 6320.50 cm^−1^, and 6320.60 cm^−1^, at 100 Torr merge into a single higher-intensity feature peaked at 6320.60 cm^−1^ at 680 Torr. The same occurs for the two spectral features which peaked at 6320.86 cm^−1^ and 6320.93 cm^−1^, resulting into a single peak at 6320.87 cm^−1^, less intense than the one at 6320.60 cm^−1^. In 2f WMS approach when the modulation amplitude is close to the absorption linewidth, the photoacoustic signal will reach the maximum. Therefore, we measured the PAS signal amplitude of 100 ppm of H_2_S in N_2_ under different pressures from 100 Torr to 680 Torr and different modulation depths. The result is shown in Figure 2.

As a representative, the H_2_S PAS spectral scans obtained for the lowest (100 Torr) and the highest (680 Torr) pressure are shown in Figure 2a. At 100 Torr, four spectral features can be identified, while the single absorption is obtained at 680 Torr as the atmospheric collisional broadening. The trend of the signal intensity of the peak occurring at 6320.60 cm^−1^ as a function of the modulation depth is reported in Figure 2b, for different system operating pressures. When the pressure was lower than 400 Torr, the trend of variation of the photoacoustic signal with the modulation depth was different from that when the pressure was higher than 400 Torr. This is due to the merging of the target absorption feature and the neighbor weaker lines under higher pressure [3]. Based on the results in Figure 2, the strongest photoacoustic signal is measured at 680 Torr and using a modulation depth of 30 mA. Therefore, these experimental conditions were selected as the ones optimizing the H_2_S PAS signal and were set for all the further measurements in this work.

To evaluate the linear response of the gold-plated photoacoustic cell-based sensor to the H_2_S concentration, different dilutions of H_2_S gas ranging from 1 ppm to 100 ppm were prepared by using the gas blender. Figure 3a shows the photoacoustic signal at different concentrations of H_2_S as well as with pure nitrogen, when the laser emission wavelength was locked to the absorption feature peak at 6320.60 cm^−1^. For each dilution, the signal amplitude was continuously measured for 200 s and recorded. Then, the signal amplitude measured at each concentration was averaged and plotted in Figure 3b as a function of the concentration. The 200 s-averaged data points were fitted with a linear regression. The R-square value after linear fitting is about 0.998, confirming the linearity of the photoacoustic sensor system response to the H_2_S concentration levels within the investigated concentration range. From the linear fit, a sensor system sensitivity of 11.39 μV/ppm is extracted.

The noise level of the sensor is defined by the standard deviations (1σ) of the signal after stabilization. The PAS signal acquired when the photoacoustic cell was filled with pure nitrogen is shown in the inset of Figure 3a and corresponds to a measured noise level of 1.55 μV. For a gas mixture of 10 ppm of H_2_S in N_2_, the signal amplitude was 110 μV leading to a signal-to-noise ratio (SNR) of 71. Therefore, with an integration time of 1 s, the minimum detection limit (1σ) was 140 ppb.

Pure N_2_ was injected into the photoacoustic cell to test the long-term stability of the H_2_S photoacoustic sensor by Allan-Werle deviation analysis. The laser was locked on the absorption line of H_2_S at 6320.60 cm^−1^, and the integration time of the lock-in amplifier was set to 1 s.

The measurement results are shown in Figure 4. It can be seen from the figure that as the integration time becomes longer and longer, the detection limit of the photoacoustic sensor can reach the lowest sub-ppt level.

After sensing performance optimization and calibration, possible spectral interference at atmospheric pressure from NH_3_, CO and CO_2_ in the sewer, whose absorption lines slightly intersect with the targeted absorption feature of H_2_S near 6230.60 cm^−1^, were investigated. With this aim, mixtures in N_2_ of 100 ppm of H_2_S and (i) 100 ppm of NH_3_, (ii) 1000 ppm of CO, (iii) 1000 ppm of CO_2_, and (iv) 100 ppm of NH_3_, 1000 ppm of CO, and 1000 ppm of CO_2_, were successively flushed into the photoacoustic cell. The measurement results are shown in Figure 5. The spectral scans in Figure 5 show that the addition of these gas species to the mixture does not affect the H_2_S peak signal at 6230.60 cm^−1^, while the slight variation of the scan shape is negligible. Therefore, we can conclude that at the concentrations reported in [24] there are negligible effects of the possible overlap of the spectral features of the typical sewer gas components, namely NH_3_, CO, and CO_2_, with the investigated H_2_S absorption feature, as well as of the possible additional energy relaxation paths on the H_2_S photoacoustic signal. In a further investigation, the higher modulation frequencies typical of quartz-enhanced PAS will be exploited to study the possible H_2_S relaxation rates variation in a matrix as complex as the sewer gas one is.

The H_2_S photoacoustic sensor based on a gold-plated photoacoustic cell was employed to measure the H_2_S content in the sewer. The gas in the sewer near the Shanxi University school canteen was measured. We connected the air pipe, filter device, pump and gas collection bag, which has a capacity of 10 L, in sequence. The filter device was used to prevent large particles in the sewer from entering the sampling bag. Then, the sewer gas in the gas collection bag was passed into the detection device through a vacuum pump. During the measurement, the pressure of the device was maintained at 680 Torr and the temperature was maintained at room temperature. The integration time of the lock-in amplifier was 1 s. The experimental results are shown in Figure 6. Based on the sensor calibration curve, it can be inferred from the figure that the H_2_S concentration of the measured sample was about 81.5 ppm, lower than the 100 ppm concentration reported by the Chinese sewer bidding document. In addition, the measured result, 85 ppm, was measured at a specific time and did not represent the average level of H_2_S gas in the school canteen.

## 3. Materials and Methods

In photoacoustic spectroscopy the sound wave signal detected by the microphone in the photoacoustic cell can be expressed by Equation (1) [25]:(1)S=C·P·α
where C is the photoacoustic cell constant, P is the incident optical power, and α is the absorption coefficient of the target gas absorption line. Therefore, we can increase the signal amplitude by increasing the photoacoustic cell constant and the light power exciting the gas molecules. For a longitudinal resonance photoacoustic cell, the photoacoustic cell constant is shown in Equation (2) [26]:(2)$$C=\frac{\left(\gamma -1\right)L^{2}Q}{\pi Vv}$$
where γ is the ratio of the specific heats at constant pressure $C_{p}$ and constant volume $C_{V}$, L is the length of the resonator, Q is the quality factor of the photoacoustic cell, V is the volume of the resonator, and v represents the speed of sound. The quality factor Q is given by [26]:(3)$$Q=\frac{R\sqrt{f}}{\sqrt{\frac{\eta }{\pi \rho }}+\left(\gamma -1\right)\sqrt{\frac{kM}{\pi \rho C_{p}}}\left(1+\frac{2R}{L}\right)}$$
where R is the radius of the resonator, η, ρ, k, and M are the viscosity coefficient, gas density, gas thermal conductivity and molar mass of the gas, respectively, f is the fundamental resonance frequency of the photoacoustic cell, which can be expressed in terms of the speed of sound:(4)$$f=\frac{v}{2L}$$

In this work, a PAS cell suitable for sewer measurements has been designed. There are many corrosive gases in the sewer, such as H_2_S, NH_3_, CO, etc. Due to the chemical nature of gold, this material is an inactive metal element. It does not react with oxygen under normal temperature or heating conditions, and can only dissolve in aqua regia, selenic acid, perchloric acid, hydrofluoric acid and nitric acid, and other corrosive substances. Such properties led to the design of a gilded photoacoustic cell, to prevent it from corrosion by acidic gases in the sewer.

The schematic diagram of the gold-plated photoacoustic cell is shown in Figure 7. The H-type photoacoustic cell consisted of a cylindrical resonator (L = 90 mm, R = 8 mm, V = 18 mL) and two identical buffer volumes at both ends. At the laser entrance port, the optical collimator (Thorlabs, USA, Model F230FC-1550) and the photoacoustic cell were fixed together through a mechanical connection. By this method, the complexity and difficulty of light path alignment can be reduced, and the light beam output by the laser can be in a straight line with the center of the cylindrical resonator. The two ends of the buffer volumes were sealed by two calcium fluoride windows, and the calcium fluoride windows were coated with anti-reflecting film, which can reduce the light absorption of the windows, thereby reducing window noise. The gas inlet and the gas outlet were designed above the buffer volumes to reduce flow noise caused by turbulence during measurement. A selected electret condenser cylindrical microphone was placed in the middle of the photoacoustic cell, where the sound pressure is the strongest [27]. A beam dump was placed at the end of the photoacoustic cell, which is fixed together with the photoacoustic cell by threads. It plays the role of absorbing excess light energy and avoids danger caused by excessive light power.

In this work, pure nitrogen is selected as the carrier gas for H_2_S detection. Nitrogen viscosity coefficient, gas density, thermal conductivity, molar mass, specific heats at constant pressure and constant volume, and speed of sound, are reported in Table 1.

At atmospheric pressure (680 Torr) and room temperature (20 °C), the theoretical photoacoustic cell quality factor Q and resonance frequency f can be calculated from Equations (3) and (4), using the molar mass and thermophysical properties summarized in Table 1, and result to be f=1938.9 Hz and Q=49. The experimental resonance curve of the photoacoustic cell is shown in Figure 8. A Lorentzian fit was used to estimate the resonance frequency and the corresponding Q value which is the ratio of the resonance frequency to the half-width of the resonance profile.

The experimental resonance frequency extracted from the fit is 1779.8 Hz, lower than the theoretical one. Such a discrepancy is generally reported in literature and is ascribed to the assumptions made to obtain Equations (2) and (3), such as one-dimensional resonator and infinite volume buffer. The quality factor of the photoacoustic cell calculated from the experimental data was Q=47, matching the theoretical one.

A schematic diagram of the photoacoustic cell-based H_2_S detection system is shown in Figure 9. A DFB diode laser with a center wavelength of 1.58 um (FITEL Inc. Model FRL15DCWD-A82) was used as the excitation light source. A function generator (Tektronix, Inc. Model AFG 3022) was controlled by a computer to generate a sine wave and a ramp wave. At the same time, the function generator outputted a TTL synchronization signal as a reference signal to the lock-in amplifier (Stanford Research Systems Model SR830). The sine wave modulated the laser to generate a photoacoustic signal. The frequency of the sine wave was 889.9 Hz, half of the resonance frequency of the photoacoustic cell, to employ the 2f wavelength modulation spectroscopy (WMS) approach. The ramp wave slowly scanned the laser wavenumber with a frequency of 10 mHz, to reconstruct the H_2_S absorption feature. The two generated waves were added and fed to the laser driver board. The temperature of the DFB laser was controlled at 31.3 °C. To enhance the photoacoustic signal (see Equation (1)), the laser power P outputted from the DFB laser was amplified to 1.4 W by an erbium-doped optical fiber amplifier (Connect Laser Technology Ltd. Model MFAS-L-EY-B-MP), which can amplify the laser beam from 30 mW to 1.5 W without changing its wavelength, and then emitted into the photoacoustic cell via optical collimator. The photoacoustic signal generated by the target gas absorbing light energy was collected by the microphone and converted into an electrical signal, which was amplified by the preamplifier and then sent to the lock-in amplifier. The lock-in amplifier then demodulated the input signal in 2f mode, i.e., at the photoacoustic cell resonance frequency. The integration time of the lock-in amplifier was set to 1 s, and the filter slope was 12 dB/oct. The lock-in amplifier transmitted the demodulated data to the computer. The data collection, processing, and logic control of the experimental system were all carried out by computer.

Measurements were performed starting from a gas cylinder containing a certified concentration of 100 ppm of H_2_S gas in nitrogen. Further dilutions were obtained by mixing the certified concentration with pure nitrogen, using a gas blender (MCQ Instruments, Model GB 100 Series). A vacuum pump (Oerlikon Leybold Vacuum Inc. Model D16C) let the gas flow through the photoacoustic sensing system and, together with two needle valves and a pressure controller (MKS Instrument Inc., Andover, MA, USA, Model 649B) was used to control the pressure within the whole gas line. A flow meter (Alicat Scientific, Inc. Model M-2SLPM-D/5 M) and the pressure meter (MKS Instrument Inc., USA, Model 649B) were used to display the gas flow rate and the real time pressure, respectively. The experimental system was controlled at room temperature, and the gas flow rate was controlled at 150 sccm by the gas blender.

## 4. Conclusions

A H_2_S photoacoustic sensing device based on a gold-plated photoacoustic cell was developed for the detection of sewer H_2_S gas. By gilding the photoacoustic cell, the detection module can be prevented from being damaged by corrosive gases in the sewer, which would further affect the detection result and stability. The diode laser power was amplified to 1.4 W by using the Er-doped fiber amplifier, to enhance the PAS signal. The pressure and modulation depth of the detection module were optimized in this article. In particular, when the pressure was atmospheric pressure and the modulation depth was 30 mA, the photoacoustic signal was the strongest. Under the conditions of atmospheric pressure and room temperature, the linear responsivity of the PAS sensor with the H_2_S concentration was verified and a sensitivity of 11.39 μV/ppm was measured. An H_2_S detection limit of 140 ppb was reached with an integration time set to 1 s. In order to demonstrate the detection capability of the device, the gas in the sewer near the Shanxi University school cafeteria was analyzed. A concentration of 81.5 ppm was measured, below the 100 ppm limit reported by the Chinese sewer bidding document.

## Figures and Tables

**Figure 1 molecules-27-06505-f001:**
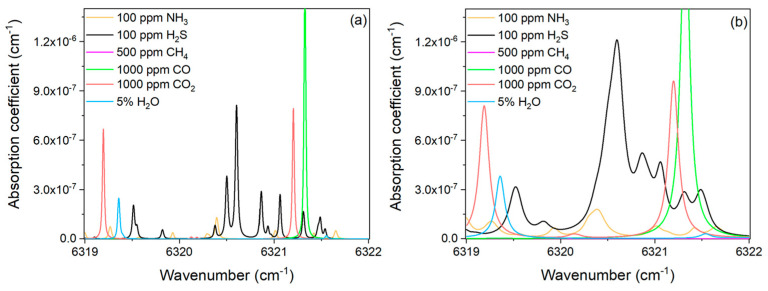
Absorption coefficient for NH_3_, H_2_S, CH_4_, CO, CO_2_, and H_2_O, based on the *HITRAN* database at 100 Torr (**a**), and 680 Torr (**b**).

**Figure 2 molecules-27-06505-f002:**
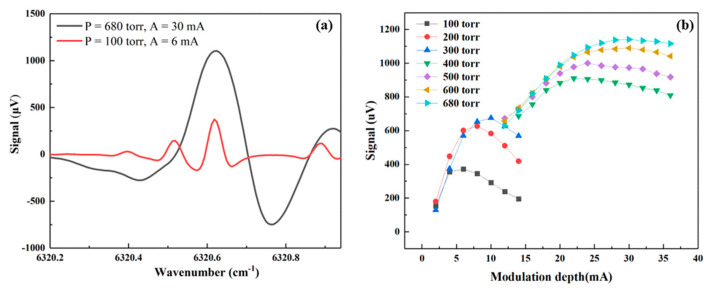
(**a**) The 2f WMS PAS signal of 100 ppm of H_2_S in N_2_ from 6320.2 cm^−1^ to 6320.95 cm^−1^ at the pressure of 100 Torr and 680 Torr. (**b**) Peak signal at 6320.6 cm^−1^ under different pressures and different current modulation depths. The experimental data are obtained at room temperature and with an integration time of 1s. The flow rate is 150 sccm.

**Figure 3 molecules-27-06505-f003:**
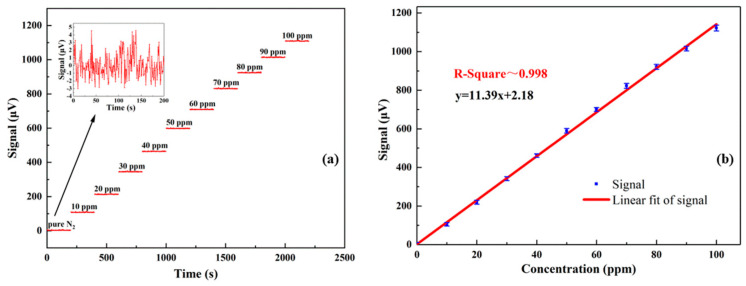
(**a**) 200 s-long measurement of the photoacoustic signal acquired for H_2_S gas concentrations from 0 to 100 ppm. Inset: photoacoustic signal acquired when pure nitrogen was flushed into the photoacoustic cell. (**b**) 200 s-averaged photoacoustic signals as a function of H_2_S concentration (blue data points) and linear fit (red solid line).

**Figure 4 molecules-27-06505-f004:**
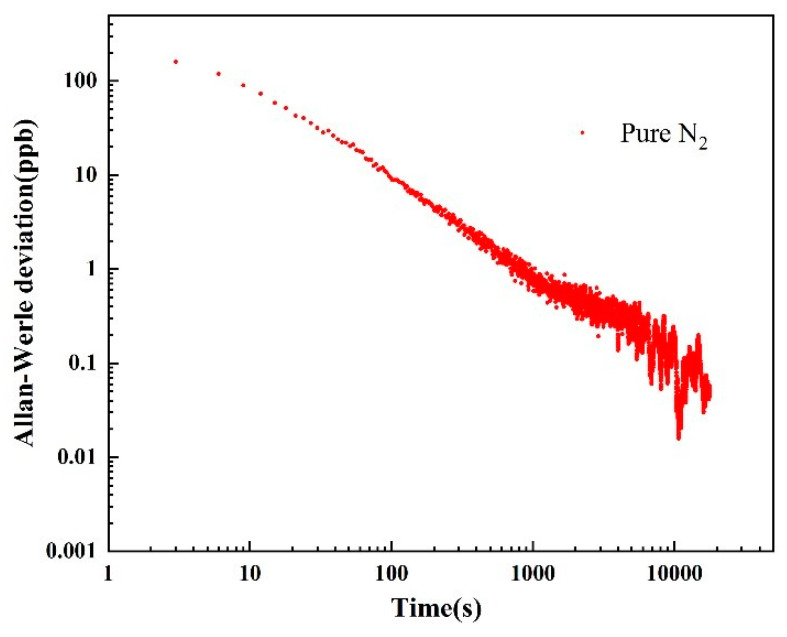
Allan-Werle deviation as a function of the integration time. The experimental data were obtained under the conditions of atmospheric pressure and room temperature.

**Figure 5 molecules-27-06505-f005:**
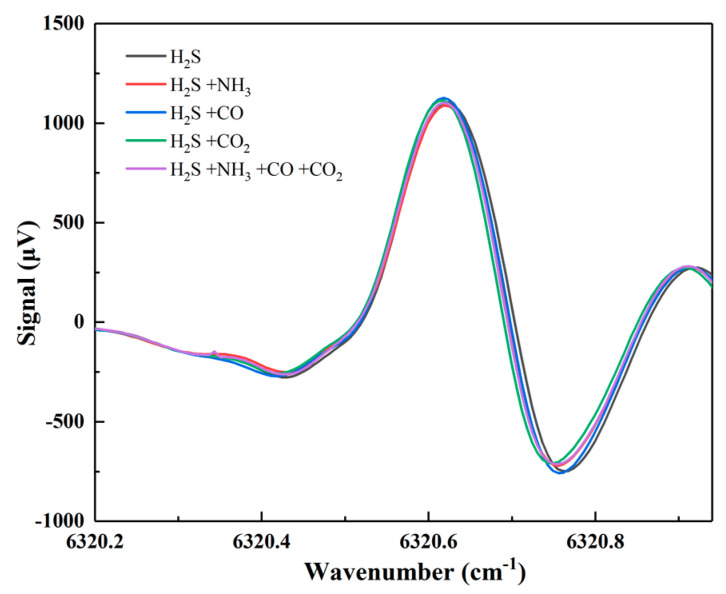
The 2f WMS PAS signal of mixtures of H_2_S, NH_3_, CO, and CO_2,_ at concentrations mimicking the sewer gas mixtures, at room temperature and atmospheric pressure.

**Figure 6 molecules-27-06505-f006:**
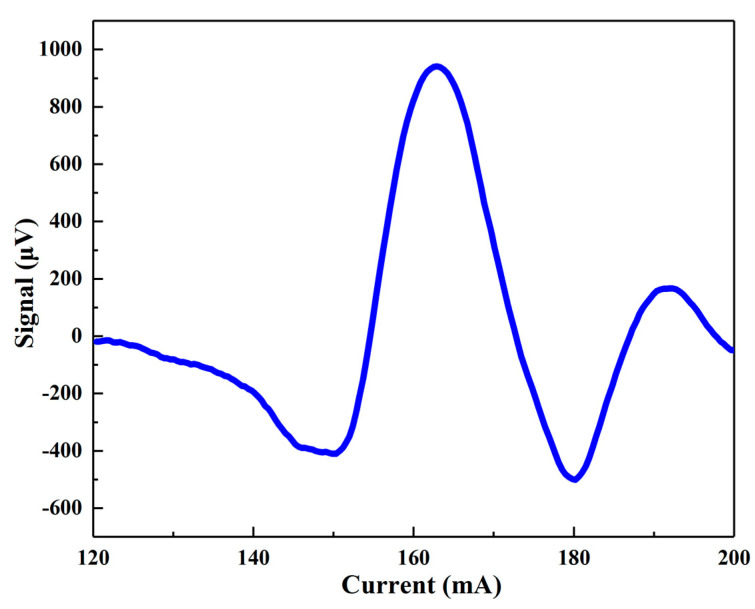
2f WMS PAS signal scan of the H_2_S absorption line in a sewer gas sample. The experiment was carried out at atmospheric pressure and room temperature. The modulation depth and excitation light power were 30 mA and 1.4 W, respectively.

**Figure 7 molecules-27-06505-f007:**
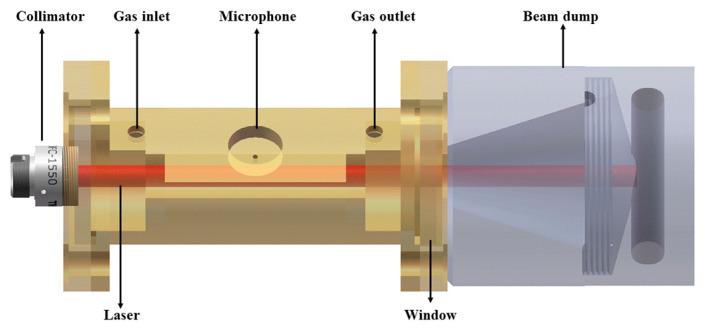
3-D model of the H-type gold-plated photoacoustic cell, the laser entrance port, and the beam dump.

**Figure 8 molecules-27-06505-f008:**
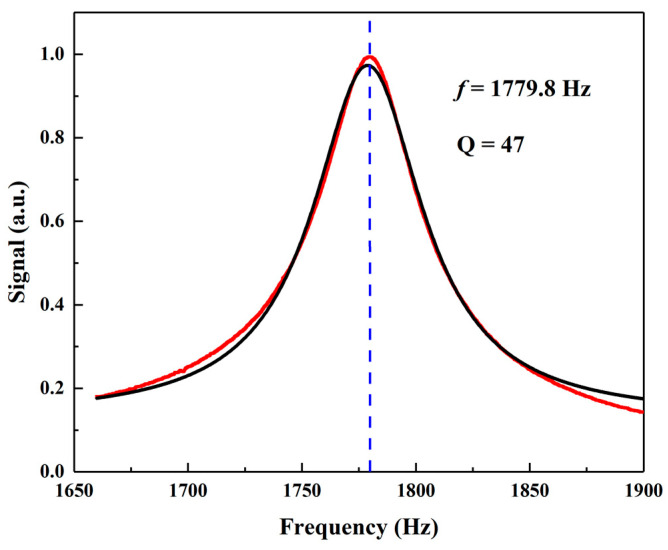
Resonance frequency response curve (red data-points) of the gold-plated photoacoustic cell in N_2_ at atmospheric pressure and room temperature and Lorentzian fit (black line). The lock-in amplifier integration time was set to 1 s.

**Figure 9 molecules-27-06505-f009:**
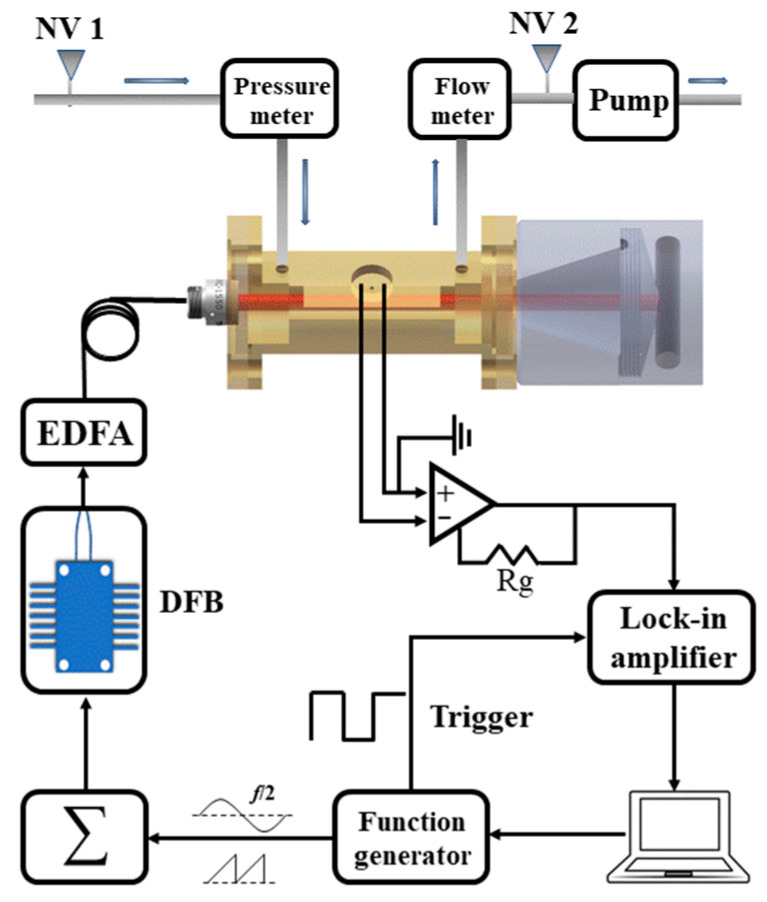
Schematic diagram of the H_2_S photoacoustic sensing system. EDFA: erbium-doped fiber amplifier; DFB: distributed feedback laser; NV: needle valve; Rg: preamplifier feed-back resistor.

**Table 1 molecules-27-06505-t001:** Molar mass M and thermophysical properties, i.e., viscosity coefficient η, density ρ, thermal conductivity k, specific heats at constant pressure $C_{p}$ and constant volume $C_{V}$, speed of sound v, of pure nitrogen at atmospheric pressure and room temperature as reported on NIST database [28].

	M	η	ρ	k	$C_{p}$	$C_{V}$	v
N_2_	28.0134 Kg/mol	17.565 uPa s	0.0372 mol/L	0.02546 W/m K	29.166 J/mol K	20.815 J/mol K	349.00 m/s

## Data Availability

All raw data are available to the corresponding author upon reasonable request.

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
