# Peer review of "Detection of Hydrogen Sulfide in Sewer Using an Erbium-Doped Fiber Amplified Diode Laser and a Gold-Plated Photoacoustic Cell"

_molecules, 2022, doi:10.3390/molecules27196505_

Round 1
Reviewer 1 Report
This work describe a photoacoustic detection module based on a gold-plated photoacoustic cell used to measure hydrogen sulfide (H2S) gas. The authors describe the sensing system, the results of calibration in laboratory and some results of gas analyses using samples collected in sewers.
The authors presented important references about the importance of sulfide gas measurement and some previous works about optical sensing system for this application. A very similar work about this application and written for the least one author that is in the current manuscript is “Near-Infrared Quartz-Enhanced Photoacoustic Sensor for H2S Detection in Biogas”, doi: 10.3390/app9245347, but it was not cited in the current manuscript.
Considering this previous review, it is not clear the novelty for this proposed work. The authors must clarify this issue.
The authors chose a not standard configuration to write the paper, where the Results and Discussion are presented after the Introduction and the Materials and Methods are presented after later. The authors should review this paper structure in order to have a more logical sequence.
Regarding the use of gas samples of sewer, the authors must describe in more details the process to collect these samples.
The authors commented in page 3, line 103: … The simulation has been 104 performed at the concentrations reported by the Chinese sewer bidding document… but this reference is not presented in the manuscript.
Author Response
We appreciate your comments and contributions to improving this manuscript. We have resolved all issues and revised the manuscript accordingly. Please see the attachment.

Reviewer 2 Report
See attached file

Author Response

(The authors gave the same response as above.)

Round 2
Reviewer 1 Report
The authors have addressed properly all the questions and comments.
Author Response
We have already addressed properly all the questions and comments in Round 1.
Reviewer 2 Report
The authors addressed several of my comments / questions. However, they did not address the, in my view, most important ones.
I suggested that the authors should address the main novelty of their work, which I did not see in the revised manuscript.
Point1: In their response the authors write non-radiative and radiative processes. Why is this not entered in to the manuscript? It is clear that the non-radiative processes are the important ones, however, the way the sentence is written they should include radiative.
Point 7: It does not matter if the excitation wavelengths are different. The molecules can still collide.
Point 8: I assume the result of the sewer measurement is the most important result? Quoting a number without an uncertainty does not give definitive conclusion.
I assume the value of 81.5 ppm is the result of a single measurement on a specific day and at a specific time? The authors can then only conclude that the value is below 100 ppm at that particular point in time. This is mentioned due to the title of the manuscript, which indicate that the paper is about detection in sewers.
